# Genomic effects of population collapse in a critically endangered ironwood tree *Ostrya rehderiana*

Yongzhi Yang[1,2], Tao Ma[1], Zefu Wang[1], Zhiqiang Lu[2], Ying Li[2], Chengxin Fu[3], Xiaoyong Chen[4], Mingshui Zhao[5], Matthew S. Olson[6] & Jianquan Liu[1,2]

Increased human activity and climate change are driving numerous tree species to endangered status, and in the worst cases extinction. Here we examine the genomic signatures of the critically endangered ironwood tree *Ostrya rehderiana* and its widespread congener *O. chinensis*. Both species have similar demographic histories prior to the Last Glacial Maximum (LGM); however, the effective population size of *O. rehderiana* continued to decrease through the last 10,000 years, whereas *O. chinensis* recovered to Pre-LGM numbers. *O. rehderiana* accumulated more deleterious mutations, but purged more severely deleterious recessive variations than in *O. chinensis*. This purging and the gradually reduced inbreeding depression together may have mitigated extinction and contributed to the possible future survival of the outcrossing *O. rehderiana*. Our findings provide critical insights into the evolutionary history of population collapse and the potential for future recovery of the endangered trees.

[1] Key Laboratory of Bio-resource and Eco-environment of Ministry of Education, College of Life Sciences, Sichuan University, 610065 Chengdu, China. [2] State Key Laboratory of Grassland Agro-Ecosystem, College of Life Sciences, Lanzhou University, 730000 Lanzhou, China. [3] Key Laboratory of Conservation Biology for Endangered Wildlife of Ministry of Education, and College of Life Sciences, Zhejiang University, 310058 Hangzhou, China. [4] School of Ecological & Environmental Sciences, East China Normal University, Dongchuan Road 500, 200241 Shanghai, China. [5] Zhejiang Tianmushan National Nature Reserve Management Bureau, 310058 Hangzhou, China. [6] Department of Biological Sciences, Texas Tech University, Box 43131Lubbock, TX 79409-3131, USA. These authors contributed equally: Yongzhi Yang, Tao Ma.  Correspondence and requests for materials should be addressed to J.L. (email: liujq@nwipb.cas.cn)

The combined effects of changing climate and human activities are assumed to have reduced species throughout the world to critically small population sizes[1–4]. For example, during the last glacial maximum (LGM), the ranges of most temperate species moved south and contracted as the temperature decreased[5,6]. Although many species recovered at the end of LGM and rapidly expanded north with Holocene climate warming, numerous species became extinct[7] or failed to recover to pre-LGM population sizes[8,9]. At the same time, human populations expanded rapidly in the Holocene[10] and negatively affected plant and animal recoveries through both hunting and land clearing[11,12]. Increasing human disturbance and climate warming in the modern era are raising further concerns of accelerated worldwide extinction rates[1,2].

Studies of the demographic history and extent of the genomic erosion leading to endangerment may both inform management and policy decisions[13] and provide a detailed empirical description of the impact of population collapse on genomic diversity and genetic load[14]. In endangered Felide, for instance, population declines predate the beginning of the Holocene[15,16], and these and subsequent bottlenecks have resulted in the accumulation of genetic load and excessive runs of homozygosity that span the genome[17,18]. Similar patterns of long-term decline are apparent in the genome of Mountain Gorillas, which caused extensive homozygosity and increased genetic load[19]. Paradoxically, this study also revealed a reduction in severely deleterious loss-of-function variants, perhaps resulting from the increased efficacy of purging of deleterious homozygous recessive alleles in small populations[20], which may have helped Mountain Gorillas survive at a low population size for thousands of generations. To date, genomic studies of critically endangered species have been conducted on animals, but plants have unique aspects to their life histories that may result in different outcomes as population size declines[3]. For instance, most plants are capable of self-pollination[19,21], which even at low levels provides a mechanism of systematic inbreeding not present in animals that will strongly contribute to purging of deleterious recessive alleles[20,22].

Trees comprise the dominant elements of the terrestrial landscape and are foundations for ecological stability and longevity in many biomes[23]. Their extinctions will inevitably lead to additional extinctions of species that rely on trees as a component of their fundamental niche. A total of 1208 trees are currently listed as critically endangered[24]; however, it remains unknown how these foundation species became endangered[1,2] and why they have survived longer than has been predicted[3]. The most effective way to increase the population size and conserve endangered trees is to directly plant wild-collected seeds or clonally propagate genotypes with stem cuttings[25], but these strategies require oversight because they may exacerbate inbreeding if parental genomes are not well represented, especially if the source population is extremely small. Here we aim to address these fundamental questions through comparing genomic patterns of diversity between the critically endangered *Ostrya rehderiana* (IUCN Red List)[24] and the widespread *O. chinensis*. *O. rehderiana* is native to southeastern China where rice was first domesticated[26,27]. This region has been heavily populated by humans for thousands of years. Although seeds from the remaining few large trees have been successfully germinated and grown to maturity[28,29], wild populations may soon be extinct. The close relative *O. chinensis* (=*O. multinervis*), however, has relatively large wild populations, which are distributed from southeastern China to the high mountains of southwestern China[30], an area that was only recently colonized by humans and remains sparsely populated[31,32]. Wood produced by both species is extremely hard and is highly prized for construction of boats and religious temples[33]. Both of these species are deciduous, monecious, primarily outcrossing, and rarely reproduce clonally[34], so factors impacting historical fluctuations in population size likely are associated more with extrinsic factors associated with the locations of their ranges than differences in life history.

We sequenced and assembled de novo genomes of both *O. chinensis* and *O. rehderiana* and re-sequenced 13 additional individuals of each species for population genomic analyses. Today only five *O. rehderiana* very old individuals (>100 years old) currently reside in a single wild population, and about 30 years ago, ~300 *O. rehderiana* trees were planted from successfully germinated seeds intentionally collected from the old stand, which at that time was comprised of six or seven mature trees[28,29,35]. Comparisons between the old and young *O. rehderiana* trees allowed us to contrast the very recent impacts of inbreeding on genomic diversity to longer-scale demographic impacts. Based on these genomic data, we addressed the following questions: (1) do these two species show similar demographic histories in response to the Quaternary climate change? If not, when did their demographics begin to diverge? (2) Have deleterious variations accumulated in the endangered species at a greater rate than in the widespread tree, and have they impacted the potential to recover? and (3) have more highly recessive deleterious variations been purged by drift in the endangered species? The answers to these questions will aid in identifying plant genomes that are on the cusp of demographic collapse.

## Results

**Genome assemblies and annotations.** De novo genomes of one 300-year-old *O. rehderiana* individual and one *O. chinensis* individual (>20 years old) from wild populations were sequenced to 128× and 340× depth of coverage (based on an estimated genome size of ~386 Mb), respectively (Supplementary Fig. 1, Supplementary Tables 1 and 2). The assembled genome sequences were 366.2 Mb (scaffold N50 of 2.31 Mb; contig N50 of 21.96 kb) in *O. rehderiana* and 371.6 Mb (scaffold N50 of 0.81 Mb; contig N50 of 13.65 kb) in *O. chinensis* with a high contiguity, coverage, and accuracy (Fig. 1, Table 1, Supplementary note 1, Supplementary Fig. 2, Supplementary Tables 3–7). In addition, both genomes contained more than 51% repetitive elements, and a total of 27,831 and 31,152 protein-coding genes were predicted in the *O. rehderiana* and *O. chinensis* genomes, respectively (Fig. 1, Table 1, Supplementary note 1, Supplementary Figs. 3 and 4, Supplementary Tables 8–13). Neither species has undergone a recent whole genome duplication (WGD; Supplementary Note 2, Supplementary Figs. 5 and 6), as found for *Betula pendula*, from another genus within the Betulaceae[36], and the divergence time between the two ironwood species was estimated at ~6.95 (3.4–12.9) million years ago (Mya) based on the fossil-calibrated phylogeny (Supplementary Fig. 7). A total of 243 unique and 526 expanded gene families were present in *O. rehderiana*, whereas 434 unique and 880 expanded gene families were present in *O. chinensis* (Supplementary Figs. 8 and 9, Supplementary Tables 14–17). We also identified 590 gene families that expanded in the ancestral lineage of the two ironwood species with predicted functions enriched in lignin catabolism, hydrolase activity, and β-galactosidase activity compared with silver birch (Supplementary Fig. 9, Supplementary Table 18). Of these gene families, 43 were related to wood formation (Supplementary note 3 and Supplementary Table 19). For example, the expanded fasciclin-like arabinogalactan protein gene family (FLAs, Supplementary Fig. 10) are critical for regulating stem strength and stiffness by affecting the molecular composition and architecture of the secondary cell wall[37].

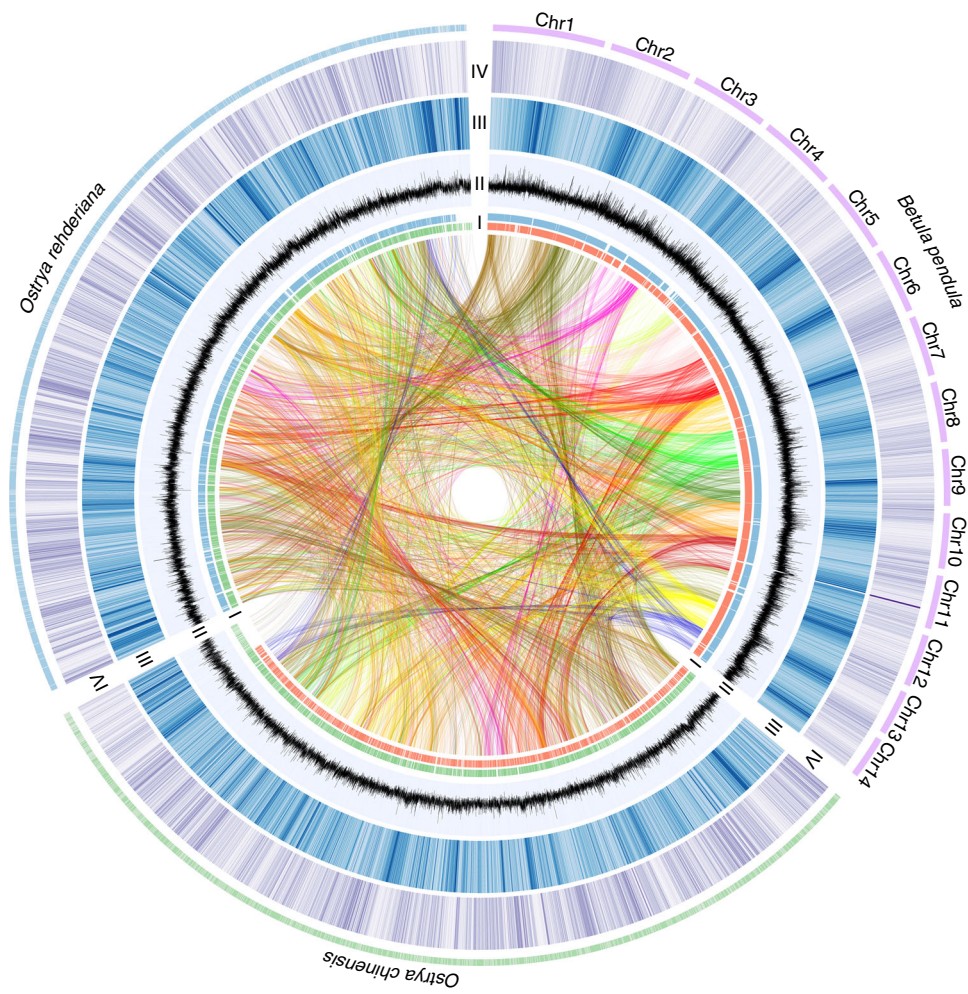

**Fig. 1** An overview of *O. rehderiana* (blue), *O. chinensis* (green), and *B. pendula* (purple) genomes. Different tracks denote (moving outwards): (I) synteny blocks between *O. rehderiana* and *O. chinensis* (green), *O. rehderiana* and *B. pendula* (blue), and *O. chinensis* and *B. pendula* (green); (II) GC content (0.2–0.8); (III) repeat density at 10 kb (0–1, from white to dark blue); and (IV) gene density in 10 kb (0–30, from white to purple). Links in the core connect syntenous genes (MCscanX results)

| Table 1 Assembly and annotation features of two genomes | | |
|---|---|---|
| **Genome features** | **O. rehderiana** | **O. chinensis** |
| Estimated genome size (Mb) | 385.90 | 386.26 |
| Assembled genome size (Mb) | 366.20 | 371.64 |
| Number of scaffolds (≥2000 bp) | 1534 | 2888 |
| Number of N50 scaffolds | 49 | 136 |
| N50 scaffold length (MB) | 2.31 | 0.81 |
| Longest scaffold (Mb) | 10.89 | 4.34 |
| GC content (%) | 35.52 | 36.12 |
| Transposable elements (%) | 50.78 | 50.77 |
| Predicted protein-coding genes | 27,831 | 31,152 |
| Gene density | 7.90 | 8.74 |
| ncRNA | | |
| miRNA | 204 | 221 |
| tRNA | 125 | 129 |
| rRNA | 566 | 552 |
| snRNA | 113 | 109 |

**Demographic histories**. Based on re-sequencing data from the assembled genome, four additional large (>100 years old), and nine additional younger *O. rehderiana* trees (<30 years old, which were planted from seeds from up to seven large parental trees with one to two of them dead in the 1990s), and 13 additional large *O. chinensis* trees (>20 years old; Supplementary Fig. 11, Supplementary Table 20), both species exhibited high $N_e \sim 1.3$ Mya followed by two sharp declines in effective population size (Fig. 2, the same demographic trends were found when only the older *O. rehderiana* trees were analyzed; Supplementary note 4, Supplementary Figs. 12 and 13, Supplementary Tables 21 and 22). The first decline occurred from 1.2 to 0.4 Mya, which coincided with the decline of the atmospheric surface air temperature (Tsurf), the escalation of the Chinese loess mass accumulation rate (MAR)[38], and the development Naynayxungla glaciation (0.8–0.50 Mya), the largest in the Qinghai-Tibet Plateau[39] (Fig. 2). The second decline occurred between 40,000 and 8000 years ago, and was initiated during the development of the LGM[40]. Following the end of LGM, however, the $N_e$ of *O. rehderiana* continued to decline to near zero, while the population size of

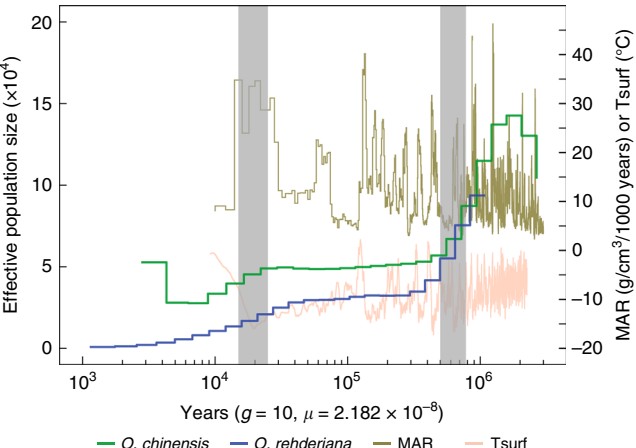

**Fig. 2** Demographic histories of *O. rehderiana* and *O. chinensis*. Shown are MSMC estimates of the effective population size ($N_e$) for *O. rehderiana* (blue, using Ore01-Ore02-Ore03-Ore04) and *O. chinensis* (green, using Och01-Och05-Och06-Och10) over the last 3 million years based on eight phased haplotypes in each species. The time scale on the *x*-axis is calculated assuming a mutation rate per generation of ($\mu$) = $2.182 \times 10^{-8}$ and a generation time of ($g$) = 10 years. The last glacial maximum (LGM) and the Naynayxungla glaciation are highlighted in gray vertical bars. MAR = loess mass accumulation rate; Tsurf = atmospheric surface air temperature

*O. chinensis* expanded in the middle Holocene (~5000 years before the present), coincident with increased temperature and precipitation in China[41]. In contrast to *O. rehderiana*, *O. chinensis* maintained a stable population size throughout the remaining Holocene and currently is not endangered.

**Accumulation of deleterious variations in *O. rehderiana*.** *O. rehderiana* displayed extremely low observed sequence diversity ($\pi = 1.66E-3$, 95% CI: $1.65E-3 - 1.67E-3$) compared to *O. chinensis* ($\pi = 2.79E-03$; 95% CI: $2.77E-3 - 2.81E-3$), *Betula pendula* ($\pi = 8.84E-3$)[36], and most other trees (Fig. 3b, Supplementary Fig. 14, Supplementary Tables 23 and 24), and is similar to *Prunus persica*, whose genome-wide diversity has been severely restricted by domestication[42]. The average observed heterozygosity and numbers of the single nucleotide variants (SNVs) across intergenic regions, introns, and coding regions were significantly lower for *O. rehderiana* than *O. chinensis* (Fig. 4a, Supplementary Tables 25 and 26). Estimates for heterozygosity in sub-samples from individuals with high sequence depth also exhibited similar lower heterozygosity in *O. rehderiana* than in *O. chinensis*, indicating that the effects of potential artifacts due to different sequencing depths were minimal (Supplementary Table 27). Among the sampled *O. rehderiana* trees, the observed heterozygosity exhibited a negative correlation with the estimated tree ages (slope = $-4.1e-6$, $r^2 = 0.68$; Fig. 3c), indicating that heterozygosity in the younger trees planted for conservation is low compared to the older trees and is consistent with the high relatedness among extant individuals (Supplementary Fig. 15). Additional patterns that are consistent with the extreme population contraction of *O. rehderiana*[43] include a more uniform site frequency spectrum (SFS), higher frequencies of derived alleles, and an elevated Tajima's *D* when compared to *O. chinensis* (Supplementary Figs. 16–18). Inbreeding, as estimated by the frequency of runs of homozygosity[44] (FROH; sum of ROH > 100 kb/genome effective length), was higher in *O. rehderiana* (FROH range 0.31–0.45) than in *O. chinensis* (FROH range 0.07–0.19; Fig. 4b); every *O. rehderiana* individual exhibited several ROH >

1 Mb, whereas the longest ROH in *O. chinensis* was <0.63 Mb (Supplementary note 5, Supplementary Fig. 19). Importantly, patterns of genomic diversity indicated that the nine young trees (<30 years old) of *O. rehderiana* were derived from only one common maternal and more than two paternal parents, which exacerbated the low levels of population genomic diversity and increased inbreeding (Supplementary note 6, Supplementary Fig. 20). Finally, *O. rehderiana* had slow linkage disequilibrium (LD) decay. Half the maximum $r^2$ was not attained until ~444 kb, whereas half the maximum $r^2$ for *O. chinensis* was attained at ~29 kb (the difference between species was not affected by including the young *O. rehderiana*, Supplementary note 6, Fig. 4c, Supplementary Fig. 21, Supplementary Table 28). A negative correlation between the population recombination rate ($\rho$) and the number of deleterious mutations, was detected in both *O. rehderiana* and *O. chinesis* (Supplementary Fig. 22), a finding consistent with previous studies[45,46].

Several patterns indicated consistently weaker purifying selection and greater accumulation of genetic load in *O. rehderiana* compared to *O. chinensis*[23,47]. The ratio of the heterozygosity of zero-fold to four-fold degenerate sites showed a negative relationship with neutral heterozygosity and was significantly elevated in *O. rehderiana* compared to *O. chinensis* (*T*-test, $P < 0.02$, Fig. 4d). The site frequency distribution for derived deleterious variants, as identified by PolyPhen2[48], PROVEAN[49], and SIFT[50], was more uniformly distributed across frequency classes in *O. rehderiana* than in *O. chinesis*, and the rare frequency classes were enriched for deleterious (DEL) and tolerated (TOL) sites compared to synonymous (SYN) sites in *O. chinesis*, but not in *O. rehderiana* (Supplementary Figs. 16 and 17), which were both indicative of weaker purifying selection in the critically endangered species. To further explore the patterns in genetic load, we estimated the proportion of the heterozygous, homozygous ancestral, and homozygous-derived mutations within four categories: SYN, TOL, DEL, and loss of function (LoF) among all samples trees (Fig. 4f) and using only the five old *O. rehderiana* trees (results were consistent throughout; Supplementary Fig. 23, Supplementary Table 29). Consistent with the higher relatedness in the extant *O. rehderiana* trees, observed homozygosity was higher in *O. rehderiana* than in *O. chinensis* for SYN, TOL, and DEL variants (all *P* values were <0.05, Mann–Whitney *U*-test). In *O. rehderiana*, an average of 5350 DEL variants were homozygous across 3793 genes in each individual, whereas in *O. chinensis* 4359 DEL variants were homozygous across 3793 genes in each individual (Supplementary Fig. 24, Supplementary Data files 1 and 2). In total, *O. rehderiana* individuals carried ~104 more derived DEL deleterious alleles than *O. chinensis*. For the LoF variants, however, *O. rehderiana* individuals carried proportionally fewer derived homozygous LoF variants than *O. chinensis* individuals (~501 homozygous LoF variants in ~485 genes for *O. rehderiana* vs. ~770 in ~691 genes for *O. chinensis*; Fig. 4f, Supplementary Fig. 24, Supplementary Table 29). GO annotations for DEL and LoF mutations were enriched in the Molecular Functions category (Supplementary note 7 and Supplementary Table 30).

In the nine young trees, the distribution of all four categories of sites (SYN, TOL, DEL, LoF) showed a lower number of heterozygous-derived sites, a slightly higher number of homozygous sites, and a similar number of total derived alleles compared to the five old trees, but not all comparisons were statistically significant (Supplementary Fig. 20d). To address whether the increased genetic load in the young *O. rehderiana* trees had detrimental effects on fitness, we counted the numbers of developed cymules after pollination (Supplementary Note 8, Supplementary Fig. 25). We found that the young trees produced fewer developed cymules per catkin than the old trees

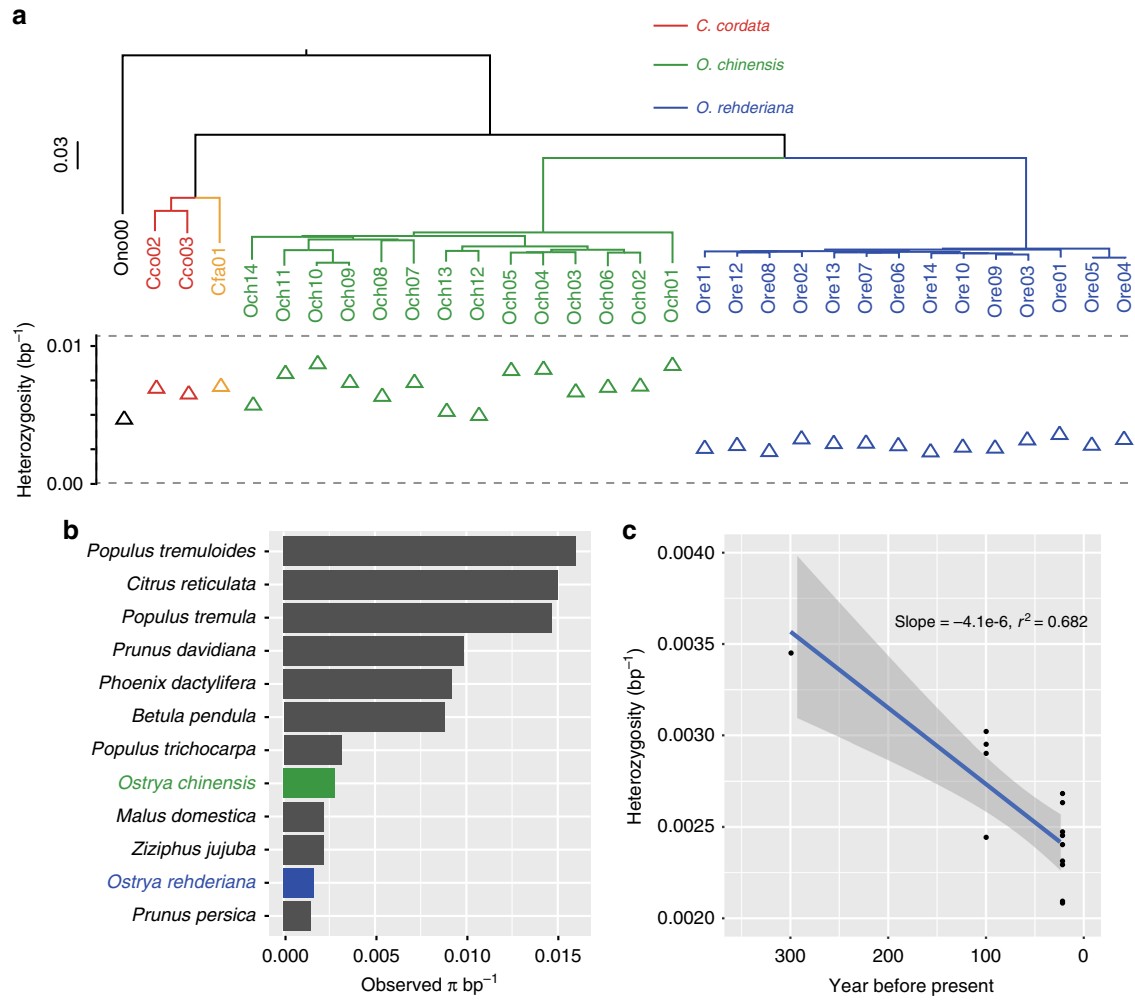

**Fig. 3** Phylogenetic and genetic diversity for *O. rehderiana* and *O. chinensis*. **a** A neighbor-joining phylogenetic tree constructed using whole-genome SNP data: *O. rehderiana* (blue), *O. chinensis* (green), *C. cordata* (red), *C. cordata* (yellow), and *O. nobilis* (black). **b** Genome-wide sequence diversity ($\pi$) for 12 tree species. **c** Genome-wide observed heterozygosity for each individual of *O. rehderiana*

(nested ANOVA, $F_{1,8} = 11.3$, $P < 0.01$), suggesting that reproductive fitness was reduced in the younger trees as expected from inbreeding depression (Supplementary note 8, Supplementary Fig. 26, Supplementary Table 31).

## Discussion

The population collapse of *O. rehderiana* was likely caused by a combination of historical climate change and anthropogenic disturbance. The population size decline in *O. rehderiana* began ~1 Mya, just prior to the Naynayxungla glaciation[39], coincident with a reduction in *O. chinensis*, but unlike *O. chinensis* the population size of *O. rehderiana* never stabilized and kept falling through the LGM and the Holocene (Fig. 2). Two factors might account for the continued decline in population size of *O. rehderiana* after the LGM. First, at some point after the LGM the $N_e$ of *O. rehderiana* may have decreased to a threshold size that constrained recovery and caused it to enter into an extinction vortex[51,52]. The low level of genetic diversity may have inhibited the adaptive potential of populations by undermining their ability to adapt to new edaphic and photoperiodic environments during migration[14] or respond to pathogens under the warming habitats of the Holocene[53]. Second, in the Holocene humans directly diminished *O. rehderiana* population sizes by cutting trees for construction and clearing land for rice farming[24,27–29]. Along with low effective population size and

impacts of genetic load, pressures from large human populations in eastern China may have tipped the balance, leading to populations that were unable to recover. Attribution of the accurate cause of the endangered status of *O. rehderiana*, however, is difficult in the absence of information on the past geographic distribution and detailed paleobotanical data.

Our study further revealed the effects of the population size decline on genome-level patterns of genetic diversity in this long-lived outcrossing tree. Trees are unusual among the angiosperms because selfing and mixed-mating breeding systems are rare compared to other angiosperms[54], suggesting that inbreeding is particularly harmful in its immediate impacts on tree populations. *O. rehderiana* exhibits high levels of inbreeding, with two of the five extant old individuals likely being monozygotic twins (Ore04 and Ore05). Although our sample size for young trees was small ($n = 9$), all of them were half-sibs, indicating that without intervention, future kin-mating among these planted trees could further enhance the inbreeding bottleneck of this endangered tree. Future management efforts should focus on reducing inbreeding, which may have already led to a decrease in female cymule development (Supplementary note 8), a likely decrease in seed set and an increase in mortality[28,29]. The observed high levels of relatedness and inbreeding among the outcrossing *O. rehderiana* trees, generated patterns of LD, runs of homozygosity,

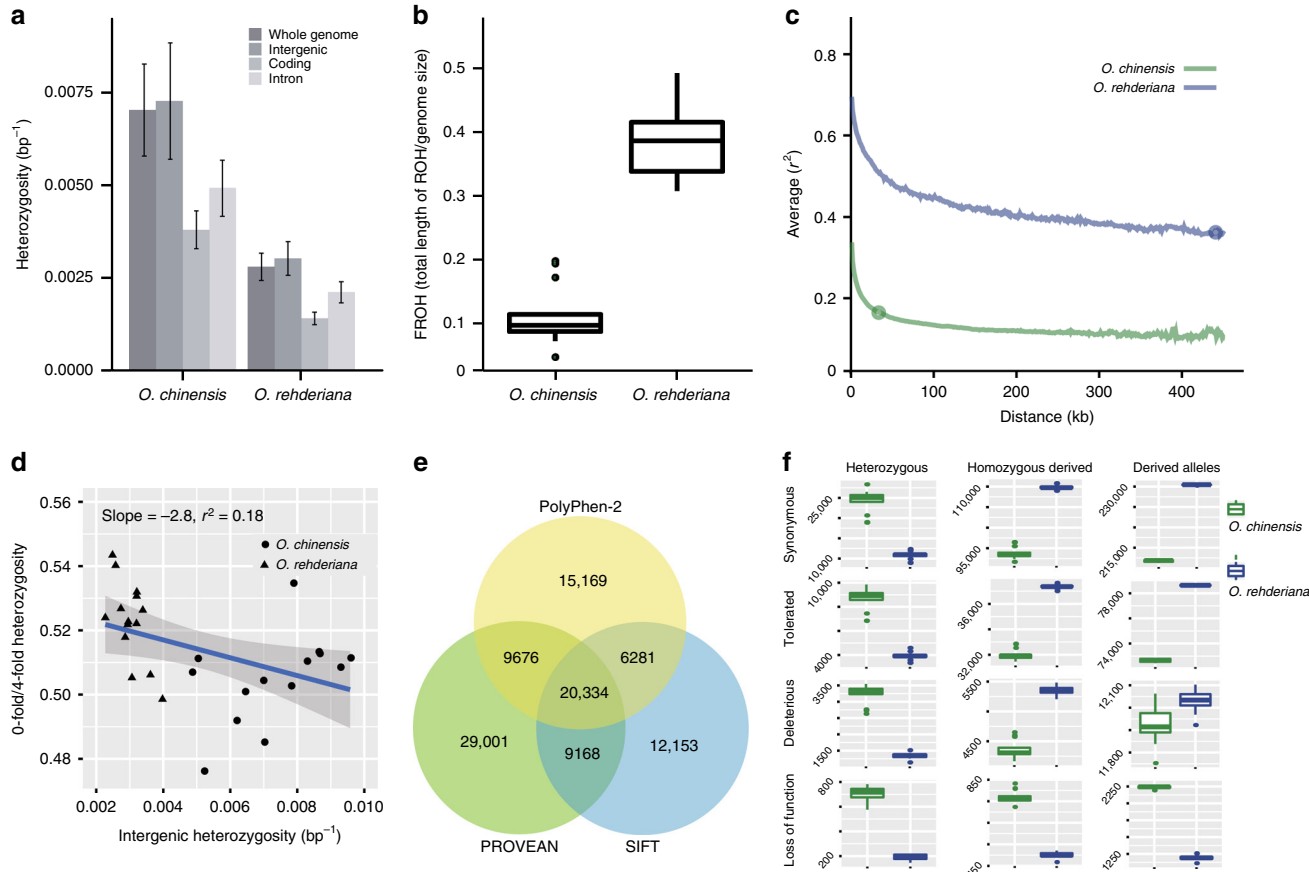

**Fig. 4** Diversity, linkage disequilibrium, and genetic load metrics for *O. rehderiana* and *O. chinensis*. **a** Bar chart of heterozygosity of the whole genome and intergenic, intron, and coding regions for each individual in the two species. Error bars represent two-fold standard deviations. **b** Box plot of FROH for each individual in the two species. The line in the center of the box represent the median values, the edges of the box represent the first and third quartiles, and the whiskers above and below the box show the range of values. **c** Linkage disequilibrium (LD) decay of the *O. rehderiana* (Ore01–Ore04) and *O. chinensis* (Och01–Och14) genomes. Open circles denote distances where the $r^2$ correlation coefficient reduces to half of its maximum (~444 kb for *O. rehderiana* and ~29 kb for *O. chinensis*). **d** The negative relationship between neutral (intergenic) diversity and the ratio of heterozygosity at zero-fold relative to four-fold sites. The triangle represents *O. rehderiana* individuals and the square represent *O. chinensis* individuals. **e** A Venn diagram of the deleterious variants predicted by PolyPhen-2, PROVEAN, and SIFT. **f** Comparison of deleterious genetic variation in *O. rehderiana* and *O. chinensis*. The total number of derived alleles is based on counting each heterozygous genotype once and each homozygous-derived genotype twice

and reduced diversity similarly observed in critically endangered animals[6–8,23,55]. In particular, genetic drift in the small populations of *O. rehderiana* has noticeably reduced the strength of purifying selection, allowing alleles with deleterious effects to persist and fix in the population and resulting in higher genetic loads than in *O. chinensis* (Fig. 4). Similar patterns have been found in other species with dramatically reduced population sizes[56,57].

In contrast to the high genetic load and reduced fitness observed in *O. rehderiana*, we found fewer LoF variants than in *O. chinensis* (Fig. 4f), possibly resulting from more effective purging of the highly deleterious variants in the extremely small populations of *O. rehderiana*. This is analogous to the proverb, 'the extremity reached, the course reversed'. Purging of the highly deleterious variants may have resulted in a gradual reduction in inbreeding depression during its long-protracted population decline (Fig. 2), which may have allowed this species to survive at low population sizes over extended time periods and may contribute to its future survival, if anthropogenic disturbance can be eliminated.

Our genomic investigation of *O. rehderiana* provides an example of the pattern of genetic diversity erosion in long-lived trees and what may be a genetic mechanism for the provisional

survival of such endangered species[3]. Although extinctions of some endangered plants lag behind the predicted date of extinction, they 'may already be functionally extinct'[5]. Therefore, many of the 1208 critically endangered trees on the IUCN Red list[24] may require unique interventions to increase both their census and effective population sizes. Future efforts should be focused on designing artificial crossing strategies to reduce inbred progenies and the loss of diversity through genetic drift, rather than increasing the total number of the surviving individuals through the collection of inbred seeds or clonal cuttings in endangered trees[25,58].

## Methods

**Source and sequencing of genomic DNA.** High molecular weight genomic DNA was extracted from mature leaves of 14 *O. rehderiana* (five large trees ≥ 100 years old, Ore01–Ore05 and nine young trees ~30 years old, Ore06–Ore14), 14 *O. chinensis* (synonym to *O. multinervis*), two *C. cordata*, one *C. fangiana*, and one *Ostryopsis nobilis* individual using the CTAB method[59]. One large *O. rehderiana* individual and one *O. chinensis* individual located in Tianmu Mountain, Zhejiang and Zehei Township, Yunnan were selected for assembling de novo genomes, respectively. High coverage shotgun-sequencing was conducted for the Illumina HiSeq2500 platform with the short, medium and long insert libraries. Statistics for the obtained reads are given in Supplementary Table 1. The additional 13 samples

for each species were re-sequenced at medium coverage (9–30×) using 500 bp insert size libraries (Supplementary Table 20).

**RNA sequencing.** Total RNAs from four tissues (roots, leaves, phloem, and xylem) were collected from the wild *O. rehderiana* and extracted using a CTAB procedure[59] for transcriptome sequencing. cDNA libraries with insert sizes of 200 bp were constructed, and sequencing was conducted using the Illumina Genome Analyzer platform. After trimming adaptor sequences and filtering out low-quality reads, the RNA data was assembled using Trinity v2.0[60] for all reads combined, and for each of the four tissues separately (See supplement methods).

**Genome assembly.** Before assembly, we filtered the low-quality sequencing reads, including removing adaptor sequences by SCYTHE (https://github.com/vsbuffalo/scythe) and trimming low-quality sequences using SICKLE (https://github.com/najoshi/sickle). We employed SOAPec[61] to minimize the influence of sequencing errors with low-frequency cutoff of 4. A total of 49.56G (or 128.39×) and 133.36G (or 345.48×) of data were retained for assembly of the *O. rehderiana* and *O. chinensis* genomes, respectively. The two *Ostrya* genomes were assembled de novo using Platanus v1.2.4[62], which is optimized for high-throughput Illumina sequence data and heterozygous diploid genomes. Briefly, Platanus assembles reads through three modules: *de Bruijn* graph-based contig assembly, scaffolding, and gap closing. The final assemblies were polished by GapCloser[61] to further close the gaps in the Platanus assemblies.

**Gene prediction.** Homology-based and de novo methods were used to predict genes in *O. chinensis* genome, whereas in *O. rehderiana* genome we also used our RNA-seq data. Integrated gene sets were generated by EVidenceModeler (EVM)[63] and functional assignments for all genes were generated by aligning their CDS to sequences available in the public protein databases including KEGG, SwissProt, TrEMBL, and InterProScan (Supplementary Table 13). The noncoding RNAs were also identified by the de novo approach based on the specific structures and the homology of the known databases (See supplement methods).

**Gene family clusters.** All protein sequences from eight species (*Arabidopsis thaliana, Carica papaya, Juglans regia, Fragaria vesca, Oryza sativa* ssp. *Japonica, P. persica, Ricinus communis, Vitis vinifera*) from NCBI were used to generate clusters of gene families (Supplementary Table 32). Gene sets were filtered by selecting the longest ORF for each gene. ORFs with premature stop codons, not multiples of 3 in length, or fewer than 50 amino acids were removed. Gene families were constructed using the OrthoMCL[64] method on the all-vs.-all *BLASTP* (*E*-value ≤ 1e−5) alignments. 1896 single-copy genes were identified within the 10 species, and subsequently used to build a phylogenetic tree. Coding DNA sequence (CDS) alignments of each single-copy family were created based on the protein alignment, using MUSCLE[65] software. The phylogenetic tree was reconstructed with PhyML[66] software under the GTR + gamma model using only four-fold degenerate sites, which are less likely to be influenced by selection and more likely to evolve in a manner consistent with a molecular clock. To estimate divergence times, the approximate likelihood calculations were conducted using the PAML[67] mcmctree program assuming a correlated molecular clock model and a REV substitution model. After a burn-in of 5,000,000 iterations, the MCMC process was performed 20,000 times with sample frequency of 5000. Convergence was checked by Tracer v1.4 (http://beast.community/tracer) and confirmed by two independent runs. The following constraints were used for time calibrations:

1. 140–150 Mya for the monocot–dicot split[68],
2. 94 Mya as the lower boundary for the *Vitis*–Eurosid split[69],
3. 54–90 Mya for *A. thaliana* and *C. papaya* split (http://www.timetree.org),
4. 90–106 Mya for *P. persica* and *J. regia* split (http://www.timetree.org).

**Gene family comparison and expansion analysis.** To study gene gain and loss, Computational Analysis of gene Family Evolution (CAFÉ)[70] software was applied to estimate the universal gene birth and death rate λ (lambda) under a random birth and death model for each branch of the phylogenetic tree using a maximum likelihood method. In addition, GO and KEGG enrichment for genes in gene families that expanded and contracted in *O. rehderiana*, *O. chinensis*, and their ancestor lineage were also calculated using GOEAST[71].

**Resequencing reads mapping.** Thirteen *O. rehderiana* and 13 *O. chinensis* individuals were selected for whole genome resequencing. Adapter sequences were trimmed from the raw reads and low-quality sequences (quality score < 20) were filtered using SCYTHE (https://github.com/vsbuffalo/scythe) and SICKLE (https://github.com/najoshi/sickle). Filtered reads were mapped to either the *O. rehderiana* or *O. chinensis* genome (scaffolds length > 2 kb) by BWA-MEM software with default parameters. Sequence alignment/map (SAM) format files were imported to SAMtools v0.1.19[72] for sorting and merging, and Picard v1.92 (http://broadinstitute.github.io/picard) was used to assign read group information containing library, lane, and sample identity. The Genome Analysis Toolkit

(GATK, v3.6)[73] was used to perform local realignment of reads to enhance the alignments near indel polymorphisms in two steps. The first step used the RealignerTargetCreator to identify regions where realignment was needed, and the second step used IndelRealigner to realign the regions found in the first step, which generated a realigned binary sequence alignment/map (BAM) file for each individual.

**SNP and genotype calling.** Single-sample SNP and genotype calling were implemented in GATK with HaplotypeCaller[73] to prevent biases in SNP calling accuracy between groups with different numbers of samples. For single-sample SNP and genotype calling, a number of filtering steps were performed to reduce false positives, including removal of (1) indels with a quality scores <30, (2) SNPs with more than two alleles, (3) SNPs at or within 5 bp from any indels, (4) SNPs with a genotyping quality scores (GQ) <10, and (5) SNPs with extremely low (<one-third average depth) or extremely high (>threefold average depth) coverage. Multi-sample SNPs were identified after merging the results of each individual by GenotypeGVCFs[73] to generate three datasets: dataset 1, SNPs from all *O. rehderiana* samples called using the *O. rehderiana* genome as reference; dataset 2, SNPs from all *O. chinensis* samples called using the *O. chinensis* genome as reference; and dataset 3, SNPs from all *O. rehderiana, O. chinensis, C. cordata, C. fangiana, O. nobilis* samples and using the *O. rehderiana* genome as reference. Each multi-sample SNP was first filtered by the GATK variant filter module with the following strict filter settings. For indels, "QD < 2.0 || FS > 200.0 || ReadPosRankSum < −20.0". For SNPs "QD < 2.0 || FS > 60.0 || MQ < 40.0 || MQRankSum < −12.5 || ReadPosRankSum < −8.0", with additional filtering steps including removing: (1) SNPs with more than two alleles; (2) SNPs at or within 5 bp from any indels; (3) genotypes with quality scores (GQ) < 10, or extremely low (<one-third average depth) or extremely high (>threefold average depth) coverage; (4) SNPs with more than two missing genotypes in either *O. rehderiana* or *O. chinensis*; and (5) SNPs showing significant deviation from Hardy–Weinberg equilibrium (*P* < 0.001) in either of the two species.

**Genome-wide genetic diversity analysis.** Genome-wide heterozygosity, within-individual SNV incidence, and SNV density within 50-kb windows were calculated in the intergenic (putative neutral, 10 kb away from coding regions) and coding regions for all 32 individuals. Of 50-kb windows, 6768 and 6456 were used with a total length of 338.4 Mb (92.4%) and 332.8 Mb (86.9%) in the *O. rehderiana* and *O. chinensis* genomes, separately. Genetic diversity (π) was calculated using 50 kb sliding windows in 10 kb steps for datasets 1 and 2 by VCFtools v0.1.12b[74]. The other detailed information for calculating FROH, LD, the population recombination rate (ρ), and population demography are shown in the supplement methods.

**Estimates of neutrality and deleterious-derived alleles.** The genome-wide ratio of heterozygosity (heterozygotes/total genotypes) of zero to four-fold degenerate sites[23,47] was calculated within coding regions based on *O. rehderiana* annotation (dataset 3). The zero and four-fold degenerate sites were identified by iterating across all four possible bases at each site along a transcript and recording the changes in the resulting amino acid. Sites were classified as zero-fold degenerate when the four different bases resulted in four different amino acids, and four-fold degenerate when no changes in amino acids were observed. Then the ratio of zero-fold to four-fold degenerate site were calculated by each individual.

Prior to detection of deleterious variants, all segregating sites in dataset 3 were phased and imputed using BEAGLE[75], and SnpEff[76] was used to classify the SNPs based on the *O. rehderiana* annotation. To avoid reference bias when identifying derived alleles and deleterious variants, we only called the polarity of variants when all three outgroups (*C. cordata, C. fangiana*, and *O. nobilis*) had identical homozygous states. Nonsynonymous SNPs were assessed using PolyPhen2 (v2.2.2r405, database: UniRef100)[48], PROVEAN (v1.1.5, database: NR)[49], and SIFT (v6.0.1, database: UniRef90)[50] with their default settings. The intersection of those three approaches may provide more accurate predictions than any single prediction approach alone, because each approach varies slightly in its prediction procedure and assumptions[51].

**Code availability.** The custom scripts have deposited in GitHub (https://github.com/yongzhiyang2012/Two_iron_wood_genome_analysis).

**Reporting summary.** Further information on experimental design is available in the Nature Research Reporting Summary linked to this article.

## Data availability
The WGS projects have been deposited at NCBI GenBank under BioProject ID PRJNA428013 for *O. rehderiana* and BioProject ID PRJNA428014 for *O. chinensis*. The genomic sequencing data and transcriptomic raw data have been deposited in the NCBI Sequence Read Archive (SRA) under BioProject ID PRJNA428015 and PRJNA428018, respectively.

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

## Acknowledgements

This work was supported by Science Foundation of China (31590821, 41471042, 91731301, and 31561123001), National Key Research and Development Program of China (2017YFC0505203, 2016YFD0600101, 2016YFC0503102), National High-Level Talents Special Support Plan (10 Thouand of People Plan), 985 and 211 Projects of Sichuan University and international collaboration '111' collaboration project and M.S.O. was supported by NSF grant DEB-1542599.

## Author contributions

J.L. conceived the study. Y.Y., Z.L., and M.Z. collected the materials. Z.L. and Y.L. prepared DNA and RNA for sequencing. T.M. and Y.Y. performed the genome assembly and genome annotation. J.L., T.M., and C.F. designed comparative genomics analyses. Y. Y., Z.W., and Y.L. performed comparative genomics analysis. Y.Y., Z.W., and Y.L. identified deleterious mutations. Y.Y, Z.L., M.Z. and X.C. built scaffolds and observed cymules. J.L., Y.Y., T.M., and M.S.O. wrote the manuscript with the input of all co-authors.

## Additional information

**Competing interests:** The authors declare no competing interests.

