## [Peer Review File · Nature Communications]

Reviewers' comments:

Reviewer #1 (Remarks to the Author):

My main issue with the manuscript is the high relatedness in the Ore samples. The 9 young trees sampled are all highly related and inbred due to an apparent extreme founder event in establishing the new populations. This biases the OR sample in several analyses, including the estimation of genetic load, demographic history etc. For this reason, most of the results from the Ore samples are simply a consequence of this extreme founder event, rather than a consequence of the species demographic history. Therefore I think that several of the conclusions in the manuscript are not well supported, and I think the authors should reconsider their analyses and the scope of the study. Due to the concerns about the relatedness in the Ore samples, perhaps the authors should consider omitting the popgen results and rebranding their study as a comparative genomic study given the nice results from the de novo assembly and annotation.

I have some specific concerns that I will list below.

1) The section about relatedness is inconsistent and therefore impossible to follow. This is a key problem in the analyses, so it needs to be addressed. First, the color scheme in Fig. S12 was not clear enough for me, so I cannot tell the difference between postulated MZ and 2nd degree relatives. Second, I do not see the postulated MZ relation between Ore3 and Ore4 (this pair is marked white in the plot, which means unrelated). In general there is a great deal of inconsistency between Fig. S12 and the text in SI lines 330-340, and even between lines 338-340 and lines 354-355 where different pairs are listed as being the parents of all young Ore. This makes it impossible to assess exactly how related the Ore samples are, except it is obvious that they are almost all closely related. The young samples must also be inbred, because they are offspring of related individuals. This needs to be taken into account when discussing the results from the genetic load, diversity and demographic analyses.

2) Which 4 Ore individuals were used for MSMC? Since most Ore samples are closely related, the impact of this on the demographic analyses should be mentioned. I highly doubt anything sensible can be inferred from inbred individuals about historical population size. At the very least some analyses must be done to show that the relatedness/inbreeding is not a problem for the MSMC results.

3) Inconsistencies in the inferred mutation rate. Different numbers are given in the Results, the figure legend and in the supp info. There is also some weird averaging going on in the SI text (quote from lines 350-353 in the SI):

"These values yielded an estimated mutation rate of $10.07E-10/9.79E-10$ mutations per generation per site per year for *O. rehderiana* and *O. chinensis*, respectively. We used the average mutation rate of $9.93E-9$ to represent the mutation rate of *O. rehderiana* and *O. chinensis* (Table S22)."

In addition, the authors report a much higher de novo mutation rate than the calibrated one, which may or may not be a cause for concern, but at least it should be discussed.

4) Figure 3c is one example of where the authors make unwarranted conclusions. The drop in het in Ore is of course due to the extremity of the founder effect and the fact that all young samples are highly related. Not due to any ongoing biological or natural process.

5) Although the language and structure of the text is in general OK to follow, there are some cases of improper or confusing wording. The SI in particular has several cases and should be gone through again to improve language.

Reviewer #2 (Remarks to the Author):

This paper is a cutting-edge example of how research on critically endangered species is transformed by whole genome sequencing and analysis. It answers questions that in the past could not have been addressed. Endangered species are hard to work on, and trees are hard to do genetic research on, but via whole genomic sequencing approaches these authors are able to overcome these problems. I expect that this is an early example of a whole new genre of papers.

There are a few minor issues that need to be cleared up before this paper is publishable.

Clarity is needed on the idea of "purging by drift". I think the authors may have got this wrong and I suggest that this phrase is either not used, or is more tightly defined.

Line 44-46 I can't find anything about purging by drift in the paper referenced (Allendorf et al 2010).

Line 50-52 "For instance, most plants are capable of self-pollination, which can both decrease effective population size, allowing drift to impact alleles with larger deleterious effects, and also increase the efficacy of the purging of deleterious homozygous recessive alleles via drift (ref 25)." This is a confused sentence. Ref 25 is about purging of deleterious alleles by their becoming homozygous. They can do this in two ways: selfing and small population size (both cause inbreeding). The latter is called "drift" by ref 25.

Figure 1. It seems rather arbitrary to do a comparison of the 25 largest scaffolds in each genome assembly, as we do not necessarily expect these to be homologous chromosomes. What is the aim of this comparison? What is it trying to show us? The links between syntenic genes suggest that many of the genes in *O. chinensis* are present in multiple copies in *O. rehderiana* but this may simply be because of the arbitrary subset of the genome that is depicted in this figure.

Line 201 "suggesting that inbreeding is universally harmful for the maintenance of tree populations" I would think inbreeding is universally harmful for all organisms. My guess would be that trees experience less selective pressure for selfing ability because they are very good at dispersing their pollen, and are therefore seldom limited in their mating opportunities.

Line 3 diving should be "driving"

Line 62. Second "and" seems misplaced or should perhaps be "where"

line 118 "Betula" is misspelled.

Reviewer #3 (Remarks to the Author):

This manuscript describes genome assembly and re-sequencing for two *Ostrya* species. *O. chinensis* is a widespread species, while *O. rehderiana* has a very limited range and is critically endangered. The authors selected about 14 samples from each species for re-sequencing and looked at diversity, LD, deleterious content, and N_e over time. The conclusion from these analyses is that *O. rehderiana* is depauperate of variation, has higher LD than *O. chinensis*, and has experienced population decline since the LGM. On the other hand, *O. chinensis* has seen its population rebound since the bottleneck associated with the LGM. These results suggest that *O. rehderiana* shows extensive genomic signatures of low N_e , consistent with its very small census population size. The authors do a nice job of the analyses and use modern methods for demography (MSMC) and apply multiple methods where appropriate (e.g., deleterious allele detection). The bioinformatics analyses (assembly, alignment, SNP calling) follow best practices in the field. The results are compelling and I think the paper will make a nice contribution to our understanding of the genomic consequences of population decline.

The primary concern I have is the way sampling was conducted, which diverged between the two species. For *O. chinensis*, only large, old trees were used, whereas for *O. rehderiana*, 5 old trees and 9 siblings from the same mother were used. While I understand that the small population size of *O. rehderiana* made sampling a more diverse array of old trees impossible, combining these two groups of trees (old trees and young siblings) is likely to affect the conclusions when compared with the samples for *O. chinensis*. Specifically, we expect much greater relatedness among the siblings (which include some full siblings...only two father trees and one mother). The authors note that "Four individuals from each species with a high coverage ($> 20 \times$) were selected to run

MSMC". Which individuals? Vastly different and potentially misleading results can be expected from the demographic analysis depending on whether these were the siblings or the old trees. Certainly, we expect siblings to have a much faster coalescence compared with relatively unrelated wild trees. Similarly, LD is expected to be much higher among close relatives, so the plot in Fig 4c may be misleading. If the demographic analyses were completed with the siblings or a mix of siblings and old trees, I think they need to be repeated with only the old trees. For LD, it would be informative to do this analysis with only the old trees as well. A few additional comments follow.

- Coverage for the de novo assembly of *O. rehderiana* was much lower than for *O. chinensis* (only about 120x vs 340x). This may affect the quality of the *O. rehderiana* assembly depending on how these depths were distributed among the libraries of different insert lengths.

- Related to this, and I may have missed this in the supp. methods, but what insert sizes were used for the genome assembly libraries?

- More details are needed on sequencing statistics for the re-sequencing libraries. What was the depth of coverage for each sample and did this vary between species? This may impact the ability to call heterozygotes and therefore affect downstream analyses.

- What is the relative level of viable seed for each species? Do these 9 siblings from *O. rehderiana* represent relatively rare healthy seeds? If so, then the results likely underestimate the fitness consequences of inbreeding in this species and this issue should be discussed.

- This is a little trivial, but I might put the species name (*O. rehderiana*) in the title, since ironwood can refer to a lot of different genera/families.

Reviewers' comments:

Reviewer #1 (Remarks to the Author):

My main issue with the manuscript is the high relatedness in the Ore samples. The 9 young trees sampled are all highly related and inbred due to an apparent extreme founder event in establishing the new populations. This biases the OR sample in several analyses, including the estimation of genetic load, demographic history etc. For this reason, most of the results from the Ore samples are simply a consequence of this extreme founder event, rather than a consequence of the species demographic history. Therefore I think that several of the conclusions in the manuscript are not well supported, and I think the authors should reconsider their analyses and the scope of the study. Due to the concerns about the relatedness in the Ore samples, perhaps the authors should consider omitting the popgen results and rebranding their study as a comparative genomic study given the nice results from the de novo assembly and annotation.

Reply: We appreciate the reviewer's comments that could help us improve our manuscript. As reviewer concerned that 9 young trees may affect the estimation of genetic diversity, LD, genetic load and population demography, we deleted all of these young trees and one of twins in the wild population of *O. rehderiana* and reanalyzed and compared all parameters between this species and widespread *O. chinensis*. We found the same trends and all comparisons have the same conclusions. We have added a section in the supplementary files to describe all comparison based on the dataset with the reduced number of the samples (Figs. S13, S23, S24, Table S25, S26; Supplementary Note 5 lines447-464). We also revised corresponding contents in the main text (lines133, 155-158).

I have some specific concerns that I will list below.

1) The section about relatedness is inconsistent and therefore impossible to follow. This is a key problem in the analyses, so it needs to be addressed. First, the color scheme in Fig. S12 was not clear enough for me, so I cannot tell the difference between postulated MZ and 2nd degree relatives. Second, I do not see the postulated MZ relation between Ore3 and Ore4 (this pair is marked white in the plot, which means unrelated). In general there is a great deal of inconsistency between Fig. S12 and the text in SI lines 330-340, and even between lines 338-340 and lines 354-355 where different pairs are listed as being the parents of all young Ore. This makes it impossible to assess exactly how related the Ore samples are, except it is obvious that they are almost all closely related. The young samples must also be inbred, because they are offspring of related individuals. This needs to be taken into account when discussing the results from the genetic load, diversity and demographic analyses.

Reply: We clarified this confusion from the following three aspects. First, we have changed the color scheme with a more distinct color for different type of the relationships. Second, the MZ relation between the wild *O. rehderiana* trees are Ore04 and Ore05, we have corrected this (see Supplementary Note 4 lines397-399). Finally, we reanalyzed and added a section to describe all the comparison with only the wild trees in detail as mentioned above (see Supplementary Note 5 lines447-464).

2) Which 4 Ore individuals were used for MSMC? Since most Ore samples are closely related, the

impact of this on the demographic analyses should be mentioned. I highly doubt anything sensible can be inferred from inbred individuals about historical population size. At the very least some analyses must be done to show that the relatedness/inbreeding is not a problem for the MSMC results.

Reply: We have added detailed information for MSMC analyses (see Fig. 2 legend, Supplementary lines 272-273). To avoid the likely impacts of the relatedness, we compared the following combinations: 1. eight haplotypes of only four wild trees when one of natural twins (Ore4 or Ore5, respectively) was excluded; 2. six haplotypes from three wild trees after deleting the related trees closer than 3rd-degree inferred by King software and 3. four haplotypes from only two wild trees after deleting the related ones closer than 3rd-degree inferred by King software. All combinations showed a similar population demography trajectories. These analyses suggested that including the related individuals did not significantly affect the simple population history this species. We added all of these analyses in the Fig. S13 and Supplementary Note 5 lines 447-464).

3) Inconsistencies in the inferred mutation rate. Different numbers are given in the Results, the figure legend and in the supp info. There is also some weird averaging going on in the SI text (quote from lines 350-353 in the SI): "These values yielded an estimated mutation rate of $10.07E-10/9.79E-10$ mutations per generation per site per year for *O. rehderiana* and *O. chinensis*, respectively. We used the average mutation rate of $9.93E-9$ to represent the mutation rate of *O. rehderiana* and *O. chinensis* (Table S22)." In addition, the authors report a much higher *de novo* mutation rate than the calibrated one, which may or may not be a cause for concern, but at least it should be discussed.

Reply: In supplementary notes 4, we added detailed information about the methods for inferring mutation rate and reanalyzed the MSMC analyses. The difference of *de novo* mutation rate and the calibrated mutation rate is a common phenomenon, and we added a paragraph to discuss this (see supplementary notes 5 lines 447-464)

4) Figure 3c is one example of where the authors make unwarranted conclusions. The drop in het in Ore is of course due to the extremity of the founder effect and the fact that all young samples are highly related. Not due to any ongoing biological or natural process.

Reply: We partially agree with this comment, but also feel that our conclusions were not "unwarranted." However, in our statement we clearly state that the "decline in the heterozygosity [was] in the younger trees planted for conservation." Note that these trees were plants from wild-collected seeds and were not the result of the controlled crosses. The relatedness between the nine young *O. rehderiana* individuals were not only within the 1st-degree, but also included trees with 2nd-, 3rd-degree. Although they were artificially planted, they do represent a selection of genotypes that could grow if they were able to establish from the older trees. Given that there were only four independent genotypes (Ore4 & Ore5 are likely twins) derived from the older population, even if the progeny resulted from multiple crosses, there is no other possibility that young samples could not be inbred.

5) Although the language and structure of the text is in general OK to follow, there are some cases of improper or confusing wording. The SI in particular has several cases and should be gone through again to improve language.

Reply: One English native speaker, an associate professor from a US university (also as the main author of the ms) co-wrote the total manuscript and the supplementary notes.

Reviewer #2 (Remarks to the Author):

This paper is a cutting-edge example of how research on critically endangered species is transformed by whole genome sequencing and analysis. It answers questions that in the past could not have been addressed. Endangered species are hard to work on, and trees are hard to do genetic research on, but via whole genomic sequencing approaches these authors are able to overcome these problems. I expect that this is an early example of a whole new genre of papers.

Reply: We appreciate the reviewer's positive comments on this work.

There are a few minor issues that need to be cleared up before this paper is publishable.

Clarity is needed on the idea of "purging by drift". I think the authors may have got this wrong and I suggest that this phrase is either not used, or is more tightly defined. Line 44-46 I can't find anything about purging by drift in the paper referenced (Allendorf et al 2010).

Reply: We apologize for the confusion resulting from the unclear statements in the manuscript. We have revised these on lines 44-45 to clarify the ideas. We also comment on the general concepts of purging by drift and purging by inbreeding below.

Line 50-52 "For instance, most plants are capable of self-pollination, which can both decrease effective population size, allowing drift to impact alleles with larger deleterious effects, and also increase the efficacy of the purging of deleterious homozygous recessive alleles via drift (ref 25)." This is a confused sentence. Ref 25 is about purging of deleterious alleles by their becoming homozygotic. They can do this in two ways: selfing and small population size (both cause inbreeding). The latter is called "drift" by ref 25.

Reply: Thank you for pointing out that this sentence was confusing. It has been revised.

Figure 1. It seems rather arbitrary to do a comparison of the 25 largest scaffolds in each genome assembly, as we do not necessarily expect these to be homologous chromosomes. What is the aim of this comparison? What is it trying to show us? The links between syntenic genes suggest that many of the genes in *O. chinensis* are present in multiple copies in *O. rehderiana* but this may simply be because of the arbitrary subset of the genome that is depicted in this figure.

Reply: We have changed the Fig. 1 with three genome comparison among *O. rehderiana*, *O. chinensis* and *B. pendula*, which can display the high quality of our assemblies.

Line 201 "suggesting that inbreeding is universally harmful for the maintenance of tree populations" I would think inbreeding is universally harmful for all organisms. My guess would be that trees experience less selective pressure for selfing ability because they are very good at dispersing their pollen, and are therefore seldom limited in their mating opportunities.

Reply: We realize that "universally harmful" was probably a poor choice of words and have replaced it with "inbreeding is particularly harmful for the maintenance of tree populations" We were attempting to imply that inbreeding is not always consistently harmful in angiosperm species. Many plants are primarily selfing and these species have often purged most deleterious genes and therefore softened inbreeding depression (see review by Winn et al. Evolution 2011 65:3339).

Line 3 diving should be "driving"

Reply: Done.

Line 62. Second "and" seems misplaced or should perhaps be "where"

Reply: Done.

line 118 "Betula" is misspelled.

Reply: Done.

Reviewer #3 (Remarks to the Author):

This manuscript describes genome assembly and re-sequencing for two *Ostrea* species. *O. chinensis* is a widespread species, while *O. rehderiana* has a very limited range and is critically endangered. The authors selected about 14 samples from each species for re-sequencing and looked at diversity, LD, deleterious content, and N_e over time. The conclusion from these analyses is that *O. rehderiana* is depauperate of variation, has higher LD than *O. chinensis*, and has experienced population decline since the LGM. On the other hand, *O. chinensis* has seen its population rebound since the bottleneck associated with the LGM. These results suggest that *O. rehderiana* shows extensive genomic signatures of low N_e , consistent with its very small census population size. The authors do a nice job of the analyses and use modern methods for demography (MSMC) and apply multiple methods where appropriate (e.g., deleterious allele detection). The bioinformatics analyses (assembly, alignment, SNP calling) follow best practices in the field. The results are compelling and I think the paper will make a nice contribution to our understanding of the genomic consequences of population decline.

Reply: We are highly grateful for the reviewer's positive comments on our work.

The primary concern I have is the way sampling was conducted, which diverged between the two species. For *O. chinensis*, only large, old trees were used, whereas for *O. rehderiana*, 5 old trees and 9 siblings from the same mother were used. While I understand that the small population size of *O. rehderiana* made sampling a more diverse array of old trees impossible, combining these two groups of trees (old trees and young siblings) is likely to affect the conclusions when compared with the samples

for *O. chinensis*. Specifically, we expect much greater relatedness among the siblings (which include some full siblings...only two father trees and one mother).

Reply: As responses to the first reviewer's concern, we deleted all of these young trees and one of twins in the wild population of *O. rehderiana* and reanalyzed and compared all parameters between this species and widespread *O. chinensis*. We found the same trends and all comparisons showed the same conclusions. We have added a section in the supplementary files to describe all comparison based on the dataset with the reduced number of the samples (Fig. S13, Fig. S20, Fig. S21, Supplementary Note 5 lines454-471). We also revised corresponding contents in the main text (lines155-157).

The authors note that "Four individuals from each species with a high coverage ($> 20 \times$) were selected to run MSMC". Which individuals? Vastly different and potentially misleading results can be expected from the demographic analysis depending on whether these were the siblings or the old trees. Certainly, we expect siblings to have a much faster coalescence compared with relatively unrelated wild trees.

Reply: We have added the detailed information for MSMC analyses (see Supplementary Note 4 and the Fig. 2 legend). As the MSMC protocol pointed out, the MSMC model explicitly assumes a well-mixing Wright-Fisher population, and close relatedness will estimate extremely low population sizes in the most recent generations. To avoid this possible impact of the relatedness, we re-ran MSMC analyses with the following three different combinations: 1. eight haplotypes from only four wild trees after deleting one of two twins; 2. six haplotypes from only three trees after deleting those with relationship closer than 3rd-degree inferred by King software; and 3. four haplotypes from two trees after deleting those closer than 3rd-degree inferred by King software. All combinations showed a similar population demography trajectory although the population size is a little different (see Supplementary Note 5 lines 447-464, Fig. S13).

Similarly, LD is expected to be much higher among close relatives, so the plot in Fig 4c may be misleading. If the demographic analyses were completed with the siblings or a mix of siblings and old trees, I think they need to be repeated with only the old trees. For LD, it would be informative to do this analysis with only the old trees as well. A few additional comments follow.

Reply: We are grateful for the reviewer's suggestion and we changed Fig. 4c with only old trees for *O. rehderiana* (deleting Ore05 as one of the likely twins to Ore04). We further reanalyzed the LD results with four wild trees by deleting the nine young trees and one of twins from the wild population. We also estimated the LD within *O. chinensis* by randomly selecting four trees and repeated this estimate ten times. When comparing LD results with the same number of individuals, the distance at the half LD decay in *O. rehderiana* is about 444 kb and this in *O. chinensis* was 29 kb (including all *O. chinensis* individuals) or 165~314 kb (randomly selected four individuals). These results are consistent with a slower LD decay in *O. rehderiana* than in *O. chinensis*. We added all of these analyses and corresponding discussions in Fig. S21, Table S26 and Supplementary Note 5 lines447-464.

- Coverage for the de novo assembly of *O. rehderiana* was much lower than for *O. chinensis* (only about 120x vs 340x). This may affect the quality of the *O. rehderiana* assembly depending on how these depths were distributed among the libraries of different insert lengths.

Reply: We thank the reviewer for these constructive comments. We added a detailed genome assembly and evaluation process in the new supplementary note 1 (lines286-290). As the reviewer mentioned, the quality of the assembled genome will be influenced by the sequencing depth and the genome complexity. Based on the k-mer analysis, we found a higher heterozygosity in *O. chinensis* than in *O. rehderiana*, which was also suggested by the heterozygosity analysis in Figure 4a. Therefore, we had to sequence more NGS data to assemble the complicated *O. chinensis* genome. During the assembly process, the short insert libraries (insert size < 2 kb) were used to construct the contigs, which may have influenced the contig N50 and the single-base accuracy. The large insert size libraries (insert size > 2kb) were used to construct the scaffolds by linking the contigs, which may influence the scaffold N50. Only the short insert libraries were used to close the gaps in the scaffolds. We adopted a series of evaluation processes on assemblies of two genomes including the reads mapping ratio, reads mapping depth, transcripts mapping ratio, CEGMA and BUSCO evaluations (supplementary note 1, lines291-303). We added the details of these assemblies and the corresponding discussions in the revised manuscripts (see supplementary note 1). Assemblies of both genomes showed the high quality in the genome completeness, contiguity and accuracy for further analyses.

- Related to this, and I may have missed this in the supp. methods, but what insert sizes were used for the genome assembly libraries?

Reply: We are sorry for causing this confusion and the detailed assembly process was added in the method section (see the revised supplementary note 1: lines 286-290).

- More details are needed on sequencing statistics for the re-sequencing libraries. What was the depth of coverage for each sample and did this vary between species? This may impact the ability to call heterozygotes and therefore affect downstream analyses.

Reply: We have used a down-sample method by the Picard software to randomly thin the mapping bam to 10 × and/or 20 × if the individual had a depth great than that. As the depth increased, the coverage increased and more SNVs were detected. However, no matter how down-sampled depth created, more than 96.38% of the total SNVs were overlapped with the raw samples. In addition, all comparisons suggested a high quality of the SNP-calling. We also compared all samples with the specific depth of 10 ×. Similarly, the heterozygosity of *O. chinensis* was significantly higher than that of *O. rehderiana* (see Table S25 and main text lines135-137).

- What is the relative level of viable seed for each species? Do these 9 siblings from *O. rehderiana* represent relatively rare healthy seeds? If so, then the results likely underestimate the fitness consequences of inbreeding in this species and this issue should be discussed.

Reply: We hired five workers to have worked for 30 days to construct scaffolds to compare the reproductive fitness of old and young trees of *O. rehderiana*. On May 17, 2018, we examined the number of the developed cymules of each catkin for both young and old trees. We found more undeveloped cymules/catkin in young trees than in old trees. Seeds will be mature in August. More undeveloped cymules/catkin in young trees suggested more serious consequences of inbreeding depressions in them. We added these results and related discussions in the revised revision (see supplementary note 7 lines479-502 and the main text

lines189-192). The relative level of viable seeds for each species (including old and young trees in the endangered *O. rehderiana*) is unavailable to us up to now.

- This is a little trivial, but I might put the species name (*O. rehderiana*) in the title, since ironwood can refer to a lot of different genera/families.

Reply: We are thankful for the reviewer's suggestion, and put the species name in the title.

Reviewers' comments:

Reviewer #1 (Remarks to the Author):

My main previous concern was the high relatedness/inbreeding notably in Ore, and how this was affecting the popgen results. This issue has been addressed by the authors, by performing many of the analyses with different subsets of the Ore endangered population. I think the manuscript presents a number of useful analyses and conclusions that can help increase our understanding of what happens in severely bottlenecked species. I still have some concerns regarding the analyses about relatedness/inbreeding. I think the authors should clarify these issues in the manuscript

Although the authors redid their analyses by subsetting their Ore samples, it would appear that almost all Ore individuals (including the wild ones) are closely related, which makes sense in a population that only harbors 5 individuals. I am not sure how, given this fact, the authors estimate the kinship presented in Fig. S15. Given this extremely compact pedigree of remaining individuals, how is it even possible to estimate kinship degree and discriminate between 1st, 2nd, 3rd and unrelated (presumably higher than 3rd degree relatives)? Kinship estimation requires knowledge about allele frequencies in an outbred population, and this does not exist here. If the kinship cannot be reliably estimated because essentially all individuals are related and inbred, then the MSMC analysis of different degrees of kinships makes little sense. In that case it would not be surprising to see the same MSMC curve regardless of subsampling.

Furthermore, the authors conclude that all young Ore samples are offspring of either Ore1+Ore2 or Ore2+Ore3. This is inconsistent with Fig. S15b, in which not all young Ore have a 1st degree relation with either Ore1 or Ore3. I am not convinced that the full pedigree structure within the samples has been correctly inferred, nor the correct kinship. I don't understand what Fig. S15c is supposed to show, as it does not appear to be consistent with Fig. S15b, nor is a structure analyses the best way to assess relatedness.

I also don't understand the lack of really long ROH stretches, especially in the young Ore. If they are truly all offspring of close relatives (Ore1 and Ore2 are 1st degree relatives), I would expect to see some really long ROH. And I would also expect to see longer ROH in the young generation than in the old generation. There seems to be no difference.

In summary, I think there are some aspects of the kinship/relatedness of the Ore samples that are not convincingly presented.

Minor comments:

Line 153: replace "parental" with "paternal".

Line 153: replace "contributed to" with "exacerbated".

Line 178-181: I do not understand what is meant by the statement in L180-181, and how this number (104) is related to the two numbers of DEL variants mentioned in the preceding sentence.

L187: add "the" after "In".

L188: add "of" before "homozygous".

L89: add "the" before "five".

Reviewer #2 (Remarks to the Author):

This MS is improved relative to the first submission, but it lacks "finish" and many of the new modifications need some minor corrections. I have also just noticed that the MSMC analyses are not described in the Methods section. They should be.

I find the first sentence of the abstract confusing. What is meant by "demographic patterns"? Inbreeding does not lead directly to severe reductions in population size. This is a very poor starting sentence.

Line 10 "de novo genomes" should be "de novo genome assemblies"

The authors' findings are presented with too much certainty lines 10-22. This needs to be caveated with words like "we find", "our models suggest" or similar.

Line 14 "because of the further anthropogenic disturbance" How do you know this? Do you have direct evidence or is it speculation? Line 212 suggests that this is not known with certainty.

Line 21 "future provisional survival" I suggest this should be "possible future survival"

Line 57 suggest replace "even out of the predicated extinction deadline" with "despite predicted extinction"

Line 61 "because they are likely to exacerbate inbreeding in small populations" I can see this is true of clonal cuttings, but this is not obviously true of wild-collected seeds

Line 128 pendata should be pendula

Line 128-129 "therefore, an effective population size of ~1500 does not signal conservation concern in the genus *Ostrya*." I don't follow the logic here. Where did the 1500 number come from? Why is it "therefore" the case?

Line 140 "by" should be "due to"

Line 143 "continues to decline" should be "is low"

The authors' findings are presented with too much certainty in parts of the discussion. This needs to be caveated with words like "we find", "our models suggest" or similar.

Line 215 "unique" should be changed to "unusual". Unique is too strong

Line 217 "for the maintenance of" should be changed to "in its immediate impacts on"

I have just noticed that the MSMC analyses are not described in the Methods section!! They should be.

Reviewer #3 (Remarks to the Author):

The authors completed extensive re-analysis of the data per my comments and those of the first reviewer. I'm satisfied with the new results and encourage publication.

Reviewers' comments:

Reviewer #1 (Remarks to the Author):

My main previous concern was the high relatedness/inbreeding notably in Ore, and how this was affecting the popgen results. This issue has been addressed by the authors, by performing many of the analyses with different subsets of the Ore endangered population. I think the manuscript presents a number of useful analyses and conclusions that can help increase our understanding of what happens in severely bottlenecked species. I still have some concerns regarding the analyses about relatedness/inbreeding. I think the authors should clarify these issues in the manuscript

Reply: Thanks for the reviewer's recognition of our analysis and we have added more assessment to approve our results.

Although the authors redid their analyses by subsetting their Ore samples, it would appear that almost all Ore individuals (including the wild ones) are closely related, which makes sense in a population that only harbors 5 individuals. I am not sure how, given this fact, the authors estimate the kinship presented in Fig. S15. Given this extremely compact pedigree of remaining individuals, how is it even possible to estimate kinship degree and discriminate between 1st, 2nd, 3rd and unrelated (presumably higher than 3rd degree relatives)? Kinship estimation requires knowledge about allele frequencies in an outbred population, and this does not exist here. If the kinship cannot be reliably estimated because essentially all individuals are related and inbred, then the MSMC analysis of different degrees of kinships makes little sense. In that case it would not be surprising to see the same MSMC curve regardless of subsampling.

Reply: The estimation of kinship coefficient by KING-robust algorithm is performed for each pair of individuals independently, which does not rely on any prior population information. Therefore, the KING-robust algorithm is independent on estimates of population-level allele frequencies, sample composition or population structure. In the original paper describing the KING software, the authors also proved that KING-robust accurately estimated kinships when using rare-variant SNPs (even the minor allele frequency < 0.05) with very few individuals (e.g. for comparison of a single pair of individuals) and very small SNP datasets (e.g. 5k, 15k and 500k SNPs showed a similar results) (Ani Manichaikul *et al*, 2010, *Bioinformatics* 26: 2867–2873). Because of its accuracy and high-speed performance for the inference of genetic relationships, it has been widely used to estimate kinship degree between individuals based on genomic data, such as a age-related macular degeneration study of human (Lars G Fritsche *et al*, 2015, *Nature Genetics* 48(2):134-143), a conservation study of the mountain gorilla (Yali Xue *et al*, 2015, *Science* 348: 242-245) and a speciation study of the poplar species (Tao Ma *et al*, 2018, *Proceedings of the National Academy of Sciences of the United States of America* 115: E236–E243).

To further assess whether the KING software is suitable for our study, we tested another three pedigrees of *Ostryopsis davidiana* (unpublished data with strict artificial crossing experiments, family A: 2 parent + 10 offspring, family B: 2 parent + 9 offspring, family C: 2

parent + 2 offspring). This genus is closely related to *Ostrya* in the same family Betulaceae. Our real pedigree dataset showed that most Kinship coefficients were larger than 0.177 but less than 0.354, which is consistent with the 1st-degree relationship (as showed in the following picture). Our pedigree analysis of the artificially crossed parents and offspring confirmed the ability of the KING software to accurately estimate kinship.

In conclusion, the relatedness estimated in our *Ostrya* study is accurate for following analysis, and all the command and in-house scripts will be uploaded when the manuscript is to be accepted.

Furthermore, the authors conclude that all young Ore samples are offspring of either Ore1+Ore2 or Ore2+Ore3. This is inconsistent with Fig. S15b, in which not all young Ore have a 1st degree relation with either Ore1 or Ore3. I am not convinced that the full pedigree structure within the samples has been correctly inferred, nor the correct kinship.

Reply: We are highly grateful for these comments that reminded us to confirm origins of nine young trees. We failed to mention that one or two old trees were killed by a flood during

1990s. Therefore, as suggested by the full pedigree structure and kinship (Supplementary Fig. 15b), it is likely that one of the nine young trees was derived from Ore2+Ore1 and four from Ore2+Ore3, while the other parent of the remaining four young trees remains unclear although all of nine young trees shared together one parent Ore2 as suggested by the first-degree kinship. These Parent-Offspring kinships were mainly used to determine an estimation of mutation rate that is independent of our estimate from the fossil calibration. We therefore have revised our manuscript and now use only these five confirmed F1 offspring (rather than nine as used before) and their two parents, Ore2 and Ore1 or Ore3) to estimate mutation rate. All corresponding contents related to the mutation rate have been updated (Supplementary Notes 5, Page93: line273-278). We also clarified the presence of one or two additional trees in the original stand that may have served to be one of two parents of some planted trees.

I don't understand what Fig. S15c is supposed to show, as it does not appear to be consistent with Fig. S15b, nor is a structure analyses the best way to assess relatedness.

Reply: We have deleted the Supplementary Fig. 15c, which did not provide useful information in the kinship inference.

I also don't understand the lack of really long ROH stretches, especially in the young Ore. If they are truly all offspring of close relatives (Ore1 and Ore2 are 1st degree relatives), I would expect to see some really long ROH. And I would also expect to see longer ROH in the young generation than in the old generation. There seems to be no difference.

Reply: We have added the comparison of ROH between the old and offspring individuals. We considered three categories of the ROH length: all the identified ROH, ROH with the lengths longer than 500 kb, and ROH with the lengths longer than 1 Mb ROH. We compared all the wild individuals and offspring and the candidate parent (Ore01, Ore02 and Ore03) and offspring using all three ROH categories. We found that the mean length of each category of ROH were longer in offspring than that in the wild individuals, but this difference is not significant. When only consider the long ROH (larger than 500 kb or larger than 1 Mb) and only comparing between candidate parents and offspring, the difference was significant ($p < 0.05$). We have described this analysis in Supplementary note 4 (Page91: line233-243) and Supplementary Fig. 19.

We also compared ROH between five young individuals and their confirmed parents and found that ROH were significantly longer in the offspring (Supplementary note 4, Page91-92: line243-250 and Supplementary Fig. 19).

In summary, I think there are some aspects of the kinship/relatedness of the Ore samples that are not convincingly presented.

Reply: We have added more detail comparison to support our inference.

Minor comments:

Line 153: replace "parental" with "paternal".

Reply: Done.

Line 153: replace “contributed to” with “exacerbated”.

Reply: Done.

Line 178-181: I do not understand what is meant by the statement in L180-181, and how this number (104) is related to the two numbers of DEL variants mentioned in the preceding sentence.

Reply: The total DEL alleles were calculated by adding the heterozygosity DEL once (one allele) and the homozygous variants twice (two alleles). The average ~104 alleles difference was obtained by the average total DEL alleles in *O. rehderiana* minus that in *O. chinensis* (see the detail numbers of each individuals at Supplementary Table 27 (Page74-75)).

L187: add “the” after “In”.

Reply: Done.

L188: add “of” before “homozygous”.

Reply: Done.

L89: add “the” before “five”.

Reply: Done.

Reviewer #2 (Remarks to the Author):

This MS is improved relative to the first submission, but it lacks "finish" and many of the new modifications need some minor corrections. I have also just noticed that the MSMC analyses are not described in the Methods section. They should be.

Reply: We are grateful the reviewer’s positive comments and the section of the detail MSMC analyses were now detail added in the supplementary methods (Page87: line113-128). All the relevant analyzing commands/scripts will be uploaded when the manuscript is to be accepted.

I find the first sentence of the abstract confusing. What is meant by "demographic patterns"? Inbreeding does not lead directly to severe reductions in population size. This is a very poor starting sentence.

Reply: We have changed these sentences.

Line 10 "de novo genomes" should be "de novo genome assemblies"

Reply: Done.

The authors' findings are presented with too much certainty lines 10-22. This needs to be caveated with words like "we find", "our models suggest" or similar.

Reply: We have changed those descriptions to reflect less certainty.

Line 14 "because of the further anthropogenic disturbance" How do you know this? Do you have direct evidence or is it speculation? Line 212 suggests that this is not known with certainty.

Reply: We deleted this statement.

Line 21 "future provisional survival" I suggest this should be "possible future survival"

Reply: Done.

Line 57 suggest replace "even out of the predicated extinction deadline" with "despite predicted extinction"

Reply: Done.

Line 61 "because they are likely to exacerbate inbreeding in small populations" I can see this is true of clonal cuttings, but this is not obviously true of wild-collected seeds

Reply: Thank you. This phrase was clarified.

Line 128 pendata should be pendula

Reply: Done.

Line 128-129 "therefore, an effective population size of ~1500 does not signal conservation concern in the genus *Ostrya*." I don't follow the logic here. Where did the 1500 number come from? Why is it "therefore" the case?

Reply: This *Ne* number is the lowest point through demographic history of *O. chinensis*. From this lowest *Ne* number, this species can recover again. This number is highly correlated with the used mutation rate. In addition, it is here represented by the *Ne* threshold estimated from only one species. We therefore deleted this confusing inference in the revised manuscript.

Line 140 "by" should be "due to"

Reply: Done.

Line 143 "continues to decline" should be "is low"

Reply: Done.

The authors' findings are presented with too much certainty in parts of the discussion. This needs to be caveated with words like "we find", "our models suggest" or similar.

Reply: We have re-worded these phrases to reflect less certainty in our interpretation of the results.

Line 215 "unique" should be changed to "unusual". Unique is too strong

Reply: Done.

Line 217 "for the maintenance of" should be changed to "in its immediate impacts on"

Reply: Done.

Reviewer #3 (Remarks to the Author):

The authors completed extensive re-analysis of the data per my comments and those of the first reviewer. I'm satisfied with the new results and encourage publication.

Reply: We appreciate the reviewer's positive comments.